# BSF-EHR: Blockchain Security Framework for Electronic Health Records of Patients

**DOI:** 10.3390/s21082865

**Published:** 2021-04-19

**Authors:** Ibrahim Abunadi, Ramasamy Lakshmana Kumar

**Affiliations:** 1College of Computer and Information Sciences, Prince Sultan University, Riyadh 11586, Saudi Arabia; 2Hindusthan College of Engineering and Technology, Coimbatore 641032, India; research.laksha@gmail.com

**Keywords:** blockchain, EHR, security, confidentiality and storage

## Abstract

In the current epoch of smart homes and cities, personal data such as patients’ names, diseases and addresses are often violated. This is frequently associated with the safety of the electronic health records (EHRs) of patients. EHRs have numerous benefits worldwide, but at present, EHR information is subject to considerable security and privacy issues. This paper proposes a way to provide a secure solution to these issues. Previous sophisticated techniques dealing with the protection of EHRs usually make data inaccessible to patients. These techniques struggle to balance data confidentiality, patient demand and constant interaction with provider data. Blockchain technology solves the above problems since it distributes information in a transactional and decentralized manner. The usage of blockchain technology could help the health sector to balance the accessibility and privacy of EHRs. This paper proposes a blockchain security framework (BSF) to effectively and securely store and keep EHRs. It presents a safe and proficient means of acquiring medical information for doctors, patients and insurance agents while protecting the patient’s data. This work aims to examine how our proposed framework meets the security needs of doctors, patients and third parties and how the structure addresses safety and confidentiality concerns in the healthcare sector. Simulation outcomes show that this framework efficiently protects EHR data.

## 1. Introduction

Electronic health records (EHR)offer a convenient health record storage system that encourages electronic access to patients’ conventional paper medical records [1]. This technique helps patients to manage, create, control and distribute EHRs to friends, family, health care providers and other authorized information customers; furthermore, health care analysts and other such service providers can access these EHRs. Moreover, patients are able to share their EHRs with various parties throughout their life time so that the EHRs shift from one provider to another.

Patient access to EHRs is too restricted, and patients generally cannot just distribute this information to providers or researchers. Conflict between different providers, research institutes, hospitals and others also hinder efficient information delivery. Due to a lack of integrated information exchange with management, EHRs are fragmented rather than synchronized [2]. If a patient can manage his/her EHRs, health departments can benefit significantly because the next time a patient goes for treatment at another hospital, it could prevent another doctor from having to re-diagnose the patient’s previous medical history. The proposed system achieves the objective of enhancing collaboration and reliability among all institutions by obtaining assistance from the blockchain.

Blockchain was initially introduced in the Bitcoin white paper in 2008 by Satoshi Nakamoto [3]. The initiation of blockchain was regarded as a novel technological uprising, similar to the invention of the steam engine, and it has had a significant impact on society.

Formerly, numerous limitations have been imposed on the distribution of enormous EHRs due to data safety risks or the drip of patient’s personal data throughout the data transfer process. Moreover, EHRs are currently controlled by providers and hospitals, while patients do not have the right to manage their own EHRs independently. This article proposes a blockchain security framework for storing and transmitting EHR (BSF-EHR) efficiently and safely to deal with this problem. Standards established for data recording and identity management using blockchain technology and the EHR blockchain have already been configured. Moreover, the technology stores the audited evidence of all dealings in an unchangeable distributed ledger, which assures accountability with transparent information transfer. Thus, a patient can store health care and discovery data from physicians in their own EHRs, thereby decreasing medical accidents and protecting the patient’s privacy.

Blockchain offers a safe, decentralized infrastructure for the regulated distribution of patient’s EHRs in the healthcare industry and blockchain is an excellent solution for EHR and data transfer. Blockchain is a decentralized framework in which each block is consecutively linked. In health care, various parties need to collaborate on the management of individual EHR blockchain (modeled on the federal blockchain), for example, doctors, patients, and insurance agents and so on. A group of parties can provide quick authentication and expensive data procedures for all the contributing participants [4]. For example, the Gem Health Network [5], developed by using Ethereum blockchain, was built for acquiring patient data from various health care professionals, minimizing health resource waste and providing access to information on the rapid treatment of critical illnesses. EHRs recorded in the block can include name, patient ID and EHRs. Patients’ privacy should be protected in the procedure of distributing EHRs when using blockchain [6].

The rest of this paper is structured as follows. Previous EHR security using blockchain is reviewed in Section 2. The BSF-EHR framework is explained in Section 3. The results of the experiments are discussed in Section 4 and Section 5 concludes the paper.

## 2. Literature Survey

McGhin et al. [4] conducted a survey of the challenges in healthcare applications and the potential use of blockchain.

In [7], the authors proposed a technique, Ciphertext-policy attribute-based encryption (CP-ABE) for the Electronic Health Record (EHR). Ramani et al. [8] recommend a blockchain-based secure and effective data access method for the patient and the doctor. The proposed system also sustains the integrity of the system. MeDShare observes objects that receive data for suspicious use from the system. A tamper-proof manner is used to record all the activities offered on the MeDShare system [9]. Liang, Xueping, et al. [10] recommend a novel user-centric health data access method through a decentralized and permissioned blockchain to preserve privacy through a channel formation scheme.

Wang et al. [11] suggested a safe electronic health record (EHR) scheme using attribute-based cryptography and blockchain technology.

Nguyen et al. [12] proposed a new EHRs distributing structure that merges blockchain with a decentralized inter-planetary file system (IPFS) on the mobile cloud. Significantly, the authors created a dependable access control method based on the smart contract to attain safe EHR distribution between various medical providers and patients. The authors concluded that their work presented an efficient solution for dependable information transfers on the mobile cloud, even protecting vital medical data against possible risks.

Ismail et al. [13] suggested a lightweight blockchain framework for healthcare data management, which decreases computing and communication overlaps compared to the Bitcoin network, which divides network contributors into clusters and maintains one copy of the ledger per cluster. Their structure introduces the need for canals, which permit safe and secret transactions within a group of network contributors. Moreover, the authors suggested a solution to prevent the forgery that exists in the Bitcoin network. The authors showed the efficiency of their suggested framework in offering safety and confidentiality to the Bitcoin network by examining various threats. The authors also discuss how their suggested architecture copes with identified threats.

Bodkhe et al. [14] analyzed a variety of solutions using blockchain and also their compatibility with a variety of applications based on Industry 4.0. First, the authors explored current cutting-edge solutions to smart appliances’ compatibility with blockchain in different industry 4.0 appliances. The advantages and disadvantages of conventional safety solutions were also explained with regard to their countermeasures.

The authors investigated personal healthcare-associated problems inorganizations that can be pursued by blockchain and also its exclusive characteristics, which can be applied to address healthcare challenges [15]. They also reviewed previous work, giving a lot of consideration to the application of blockchain in the healthcare sector. Finally, they discussed the advantages and potential research opportunities for the blockchain-related technology used in the healthcare sector.

Wagh et al. [5] provided a comprehensive overview of blockchain technology. The authors provided a synopsis of blockchain architecture, security in blockchain and its benefits [16]. Furthermore, they explored its application in a new field, namely, the health sector. The authors also discuss how health records in the health information system could be protected using blockchain technology.

Ramani et al. [8] suggested blockchain as a shared scheme for healthcare systems. The authors proposed a safe and effective information access method based on blockchain fora patient with a physician in a particular health care setting. Their study of the program’s safety demonstrated that it maintains the organization’s integrity and can resist well-known attacks [17]. Furthermore, the implementation results illustrate the potential of the system proposed by the authors.

Information transfers from one company to another are recorded undamaged on MeDShare, which uses access control mechanisms to efficiently monitor information activities to identify companies involved in breaches of data [18].

Liang et al. [10] suggested an innovative customer-centered medical information distribution system using decentralization with authorized blockchain for defending confidentiality based on channel creation programs and to improve identity management based on a blockchain-supported association service [19,20]. The mobile appliance created to gather medical information from body sensor nodes also coordinates data for a cloud for distributing information to healthcare providers [21,22].

The authors utilized identity-based encryption (IBE) [23] and attribute-based encryption (ABE) [24] to encrypt healthcare information. They also utilized identity-based signatures(IBS) for develop digital signatures. This efficiently facilitates the organization of a scheme that does not require the introduction of various cryptographic schemes for various safety needs.

Bach et al. [25] conducted a comparative study of blockchain consensus algorithms, especially Ethereum, which utilizes a consensus protocol known as proof-of-work (PoW). This method allows the decentralized Ethereum networks to come to consensus on the order of transactions and account balances. This prevents customers from “double spending” their money, and also makes sure that the Ethereum chain is extremely hard to overwrite or attack.

## 3. Design Methodology

Electronic health records (EHR)are regulated by health centers instead of patients, making it difficult to obtain medical advice from various health centers. Thus, patients need to concentrate on restoring the management of their health details and their medical information [21,25]. The quick evolution of blockchain technology encourages population healthcare, including access to patient information and medical data. The technology offers patients access to extensive, consistent reports with free access to EHRs from treatment websites and providers. In this section, we describe the development of a blockchain security framework for EHR (BSF-EHR) with multiple authorities to meet the need for blockchain in shared EHR systems. This framework protects the privacy of patients and maintains the consistency of EHRs. Figure 1 demonstrates the difference between the traditional EHR system and the BSF-EHR system.

Figure 1 shows the novelty of the BSF-EHR system. On the left side, the traditional EHR system is illustrated. (1) The patient visits the doctor. (2) Then doctor treats the patient. (3) After treatment, the doctor uploads the EHR to the server. (4) For future use, the doctor can download the EHR.BSF-HER can be defied as a novel secure electronic health record of patients which could be privately shared by institutions or patients.

On the right side, the BSF-EHR system is illustrated. Using BSF-EHR, the patient is able to manage, download and share his/her EHRs independently. The proposed BSF-EHR framework consists of five parties: the patient, doctor, insurance agent, EHRs server and the data verifier. (1) Similar to the traditional EHR system, the patient visits the doctor. (2) Then the doctor treats that patient. (3) The EHR system server is one of the nodes in a blockchain network. It acts as a miner that collects transactions (EHRs) and organizes them into blocks. Whenever EHRs are created after treatment, all network nodes receive them and verify their validity. Then, the miner node gathers these transactions from the memory pool and begins assembling them into a block. The memory pool is a “waiting area” for transactions that each node maintains for itself. After a transaction is verified by a node, it waits inside the memory pool until it is picked up by a miner and inserted into a block. This new block will then be added to the blockchain. However, before the block can be added to the chain, the information contained in it must be verified by the miner. This happens by creating a so-called “hash”. A hash is a 256-bit number that uniquely identifies the data in the block. (4) After block creation, the miner distributes it to all the available nodes in the blockchain network (doctors and patients). (5) Also, BSF-EHR provides access control for each node based on its own blockchain concept. Through this, the patient can view their own EHR in their own blockchain and no one else can see the details. Additionally, a doctor can view the EHR of patients who have been treated by him/her in their own blockchain. A doctor can view only permission-granted patient blocks in his/her own blockchain. (6) For medical insurance claims, the doctor can share this EHR (copy of own blockchain) with an insurance agent. (7) The insurance agent can view the EHR of patients who claimed in his/her own blockchain and after approval, they can also provide the insurance amount to the patient. (8) Furthermore, the patient can send a data verification request (with a copy of their own blockchain) to the data verifier. (9) Finally, the data verifier checks whether the data are safe or not and provides verification results to the patient.

Generally, there are three distinct types of nodes—miner nodes, full nodes, and light nodes. Miner nodes can propose blocks and have the complete history of the blockchain. Full nodes have the complete history of the blockchain but cannot propose new blocks and light nodes rely on full nodes for the blockchain’s history. In the BSF-EHR system, the miner node is the EHR system server and both the doctor and patient are full nodes. The insurance agent plays the role of the light nodes. In the BSF-EHR system, a large number of doctors and patients and also insurance agents are available. Therefore, access control is necessary. Our BSF-EHR system provides access control.

In BSF-EHR, the patient can only see his own EHR details, and no one else can see the details. Furthermore, the doctor can only see the EHR of patients who have been treated by him. A doctor can only view permission-granted patient blocks. Similarly, the insurance agent can only view permission-granted patient blocks.

### 3.1. Benefits of Blockchain Technology in BSF-EHR System

Blockchain provides many benefits in the BSF-EHR system. These include
Better health records exchange;Increased data security and privacy;Validate the correctness of billing management;Empower the medical supply chain;Enhance the climate of trust in healthcare.

### 3.2. Algorithm Design

Algorithm 1 explains the BSF-EHR patient blockchain formation and block addition.
**Algorithm 1.** BSF-EHR Patient Blockchain Formation and Block Addition**Input****:**EHR Readings for a patient Pat1**Output****:**Patient Pat1 Blockchain formation and Add blocks to Patient Pat1 Blockchain**Step 1****:**EHR ← Patient Pat1 EHR Readings**Step 2****:**Public Key and Private Key ← Key Generation using RSA cryptography**Step 3****:**The patient uses the public key for encryption purposes. The private key shared with referred doctor and Insurance Agent for Decryption purpose.**Step 4****:**encryptedEHR ← Encrypt EHR based on Public Key**Step 5****:**hash ← Generate hash for encryptedEHR based on HMAC-SHA1**Step 6****:**Bilinear Map ← Generate bilinear maping for encryptedEHR with patient ID**Step 7****:**Create a Genesis block for Patient Pat1 blockchain using the patient name, password and patient ID**Step 8****:**Block ← Put encryptedEHR, hash with Bilinear Map**Step 9****:**Add this Block to Patient Pat1 Blockchain

To improve the security, a bilinear map function is used in BSF-EHR. A bilinear map is a function that merges elements of two vector spaces to yield an element of a third vector space, and is linear in each of its arguments. Matrix multiplication is an example.
V × W → Y
where, V = vector space 1, W = vector space 2 and Y = vector space 3.

Algorithm 1 takes encrypted EHR as vector space 1 (V) and patient ID as vector space 2 (W). Here, the identity-based encryption concept was used to generate a bilinear map. Algorithm 1 encrypt encrypted EHR (V) using patient ID (W) based on the identity-based encryption concept. This provides bilinear map (Y) (Step 6).

Figure 2 shows the Patient Pat1 blockchain. It contains the patient’s name, password and patient ID in the genesis block. Details of each disease treatment (encrypted EHR, a hash with a bilinear map) are added as a new block, one by one. Here the patient can only see his details, and no one else can see the details.

Algorithm 2 explains the BSF-EHR doctor and insurance agent blockchain formation and block addition.
**Algorithm 2.** BSF-EHR Doctor and Insurance Agent Blockchain Formation and Block Addition**Input****:**Referred patient block from the patient blockchain**Output****:**Doctor Doc1 Blockchain formation and Add blocks to Doctor Doc1 Blockchain and Insurance Agent IS1 Blockchain formation and Add blocks to Insurance Agent IS1 Blockchain**Step 1****:**Block ← Doctor Doc1 and Insurance Agent IS1 can download his referred patient Block from the patient blockchain using the private key.**Step 2****:**Block → Retrieve encryptedEHR, Hash with Bilinear Map from Block**Step 3****:**Decrypt encryptedEHR based on Private Key → EHR**Step 4****:**Doctor Doc1 and Insurance Agent IS1 can access this EHR**Step 5****:**Create Genesis block for Doctor Doc1 blockchain using doctor name, password and doctor ID and Create Genesis block for Insurance Agent IS1 blockchain using insurance agent name, password and insurance agent ID**Step 6****:**Block ← Put encryptedEHR, hash with Bilinear Map**Step 7****:**Add this Block to Doctor Doc1 Blockchain and Insurance Agent IS1 Blockchain**Step 9****:**Insurance Agent IS1 provides the Lump-sum payment for the treatment of covered illness. This amount is transferred to the particular patient block

Figure 3 shows the Doctor Doc1 blockchain. It contains the doctor’s name, password and doctor ID in the genesis block. Details of each disease treatment (encrypted EHR, a hash with a bilinear map) are added as a new block, one by one. A doctor can view only permission-granted patient blocks.

Figure 4 shows the Insurance Agent IS1 blockchain. It contains the insurance agent’s name, password and insurance agent ID in the genesis block. Details of each disease treatment (encrypted EHR with a hash) are added as a new block one by one. Similar to the doctor, the insurance agent can view only permission-granted patient blocks. No one can see the EHR because all the EHR records are in an encrypted format at blocks in the blockchain. Finally, a data verifier can check whether any patient’s blockchain is safe or not. Algorithm 3 explains the BSF-EHR blockchain validation algorithm.
**Algorithm 3.** BSF-EHR Blockchain Validation Algorithm**Input****:**Patient Pat1 blockchain**Output****:**Safe or not**Step 1****:**Blockchain BC ← Download Blockchain of Patient Pat1**Step 2**
BlockchainStatus = “Safe”**Step 3****:**FOR each Block B from BC**Step 4****:**Block → Retrieve encryptedEHR, hash with Bilinear Map from the block**Step 5****:**newHash ← Generate new hash for encryptedEHR based on HMAC-SHA1**Step 6****:**newBilinearMap ← Generate new bilinear maping for encryptedEHR with patient ID**Step 7****:**IF ((hash == New Hash)&(Bilinear Map = newBilinearMap))**Step 8****:**Block is safe**Step 9****:**ELSE**Step 10****:**Block is not safe

BlockchainStatus = “Not Safe”.**Step 11****:**Break**Step 12****:**END FOR

## 4. Results and Discussion

This study aims to protect patients’ confidentiality and keep the consistency of EHRs by using the BSF-EHR framework. This experiment used a blockchain that was created using POJO in Java. Blockchain is a technology that facilitates transactions between non-trusted parties. Here, a blockchain is a set of blocks containing one or more transactions. Each block is hashed, and then the hashes are linked, hashed, reconnected, and re-hashed until there is a hash, which is the Merkle root of a Merkle tree. Every block records the hash of a former block by connecting the blocks jointly. This guarantees that a block cannot change without changing all of the subsequent blocks. This experiment holds data as a string, containing Ethereum style smart contracts. This experiment investigated the BSF-EHRs framework and calculated the effectiveness of the BSF-EHRs sharing framework through two primary evaluation metrics: access control with time consumption.

### 4.1. Access Control

This experiment provides two cases, unauthorized access and authorized access, to assess the BSF-EHR framework’s effectiveness with designed access control, as demonstrated in Figure 5. The goal of the BSF-EHR framework is to permit permission-granted parties (such as the doctor and insurance agent) to extract EHRs efficiently on the blockchain while avoiding unauthorized access to EHRs resources.

Figure 5a shows the unauthorized user cannot access EHR, and Figure 5b shows the authorized user can access EHR. The outcome of the examined access control shows that BSF-EHR attains safe information distribution between customers. Most importantly, the BSF-EHR access control system is capable of protecting important EHRs against external attacks.

### 4.2. Time Consumption

This experiment evaluated the performance of the BSF-EHR framework based on the time consumption for accessing EHRs in blockchain versus centralized storage, as shown in Figure 6.

This experiment calculated the time consumption for the EHRs access procedure, from requesting data to receiving data.

The time consumption calculation process was as follows. In centralized storage, EHRs are stored in a centralized server. If a patient wants to access his/her EHR, he/she creates the EHR request. Then, we note the current time (T1), and he/she sends the EHR request to the centralized server. After receiving the EHR request from the patient, the centralized server searches and obtains his/her EHR and transmits it to the patient. Then we note the current time (T2). Therefore, the time consumption for accessing EHR = (T2 − T1) seconds.

Furthermore, time consumption is directly proportional to EHR size. If the EHR is large, the time taken to access the EHR is significant. Conversely, if the EHR is small, the time taken to access EHR is small. The time consumption varies based on the EHR size.

In BSF-EHR, each patient maintains his/her blockchain. This blockchain contains his/her EHR in an encrypted format. After decryption, he/she can access his/her EHR. Compared with centralized storage, this takes less time.

The experiment outcomes concerning various EHRs file sizes demonstrate that BSF-EHR outperforms centralized storage in terms of time consumption. This result further shows the effectiveness of the BSF-EHR framework.

Table 1 demonstrates the comparison of BSF-EHR with several related works in regard to various technical features, using two options: Y—yes (available) and N—no (unavailable). The comparison outcomes show that the BSF-EHR performs better than these related works and it offers a promising solution for enhancing present e-health applications.

## 5. Conclusions and Future Work

This paper proposes a blockchain security framework (BSF-EHR) to store EHRs effectively and securely. This framework offers patients access to extensive, consistent records and free access to EHRs, and it also protects the privacy of patients and maintains the consistency of EHRs. By using BSF-EHR, the patient is able to independently manage, download and share his EHRs. The experimental results show that BSF-EHR achieves secure data sharing between users. Most importantly, the BSF-EHR access control system is capable of protecting important EHRs against external attacks. Furthermore, the BSF-EHR framework attains lightweight EHR sharing with minimum time consumption compared to traditional centralized storage.

This blockchain security framework was only applied in healthcare domain in this study. In future, we will apply this framework to various domains such as the supply chain, Internet of Things, education, logistics, finance, banking, agriculture and accounting.

## Figures and Tables

**Figure 1 sensors-21-02865-f001:**
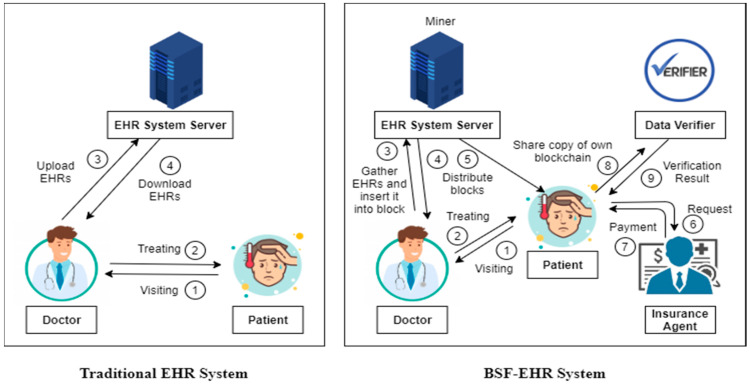
Novelty of BSF-EHR system.

**Figure 2 sensors-21-02865-f002:**
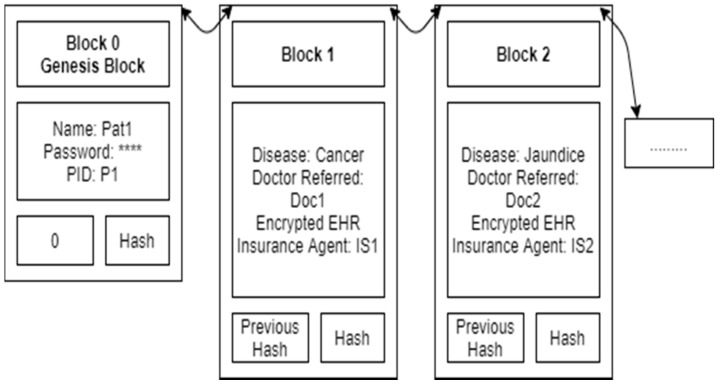
Patient Pat1 blockchain.

**Figure 3 sensors-21-02865-f003:**
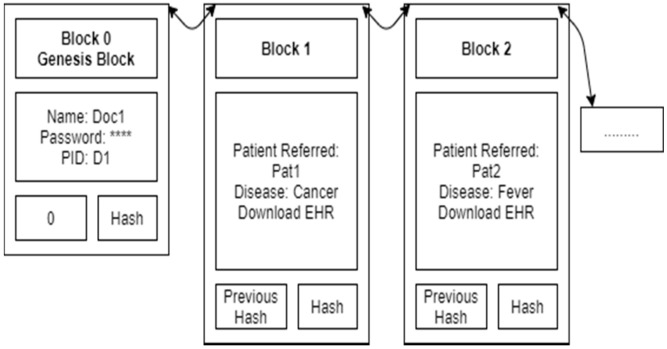
Doctor Doc1 blockchain.

**Figure 4 sensors-21-02865-f004:**
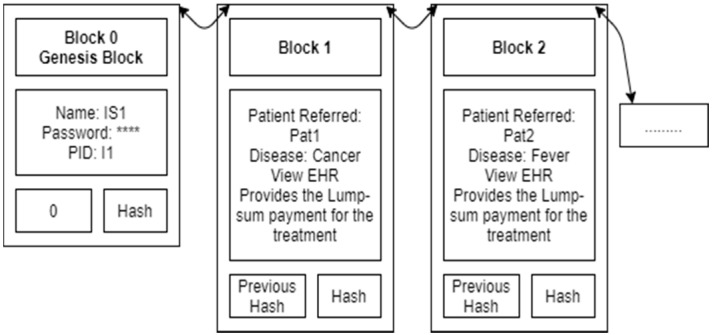
Insurance Agent IS1 blockchain.

**Figure 5 sensors-21-02865-f005:**
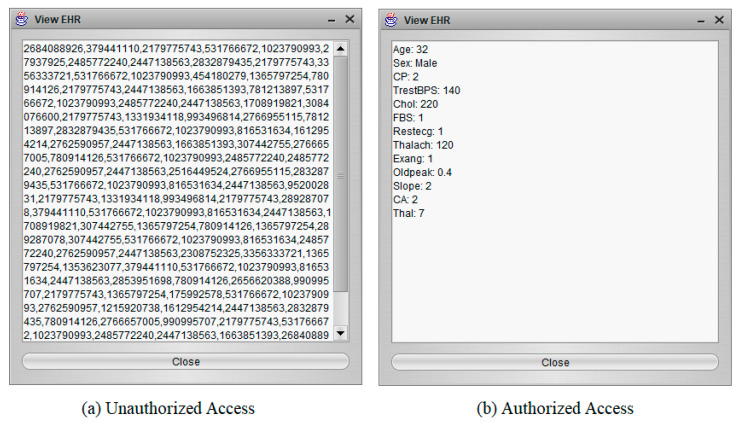
EHRs as a result of unauthorized access vs. authorized access.

**Figure 6 sensors-21-02865-f006:**
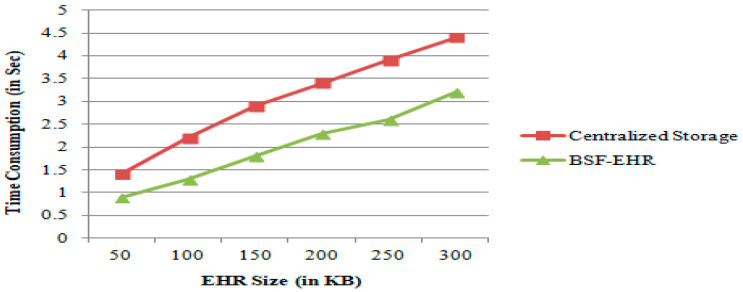
Time consumption comparison based on centralized storage vs. BSF-EHR.

**Table 1 sensors-21-02865-t001:** Comparison of BSF-EHR with a few related works.

Feature	[7]	[8]	[9]	[10]	BSF-EHR
Decentralized access	N	Y	Y	Y	Y
User authentication	Y	Y	Y	N	Y
Identity management	Y	N	Y	N	Y
Data privacy	Y	Y	Y	Y	Y
Flexibility	N	N	N	Y	Y
Availability	N	N	Y	Y	Y
Integrity	Y	Y	Y	Y	Y

## Data Availability

Not applicable.

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
