# Peer review of "BSF-EHR: Blockchain Security Framework for Electronic Health Records of Patients"

_sensors, 2021, doi:10.3390/s21082865_

Round 1

Reviewer 1 Report

The topic of the article is very topical - the availability of patient data from various doctors / hospitals is still a big problem. The article does not state whether the proposed system was tested somewhere in real operation. If so, what is the motivation of hospitals to provide this data? This may vary from country to country depending on the legislation. How will the export of data from different medical entities be solved - I assume that these can have different data formats and technologies.

The method used is clearly described in the article, but I lack a closer explanation of the technological context. What steps would a patient - potential person interested in using the proposed framework - have to take in order to use this framework? Maybe add some process diagram?

The framework's own architecture (software solution) could also be described more. The use cases listed in Chapter 4 do not state what this applies to. Is it output from the BSF or is it from an application that communicates with the BSF?

Author Response

As per the reviewer comments, we have addressed all the points and given them as an attachment. 

Reviewer 2 Report

[BFR-EHR]
- It has been mentioned that the BSF-EHR access control system can protect the EHR from external attacks.
Since the proposed system only encrypts EHR, it is necessary to consider whether there is a way to deal with it when attacking the server from outside.
- Like the preceding, the proposed framework consists of 5 parties, but the EHR server has most of the data.
There is not enough evidence to support it as a secure system.
- It is necessary to consider scalability constraints in terms of the tradeoff between the transaction volume and  the computer power for processing the transactions.
- When stored health data needs to be deleted from the blockchain, it directly conflicts with the invariant objective of the blockchain support solution. You need to think about how you can solve it.

[Replacement of Figures and Letters]
- Algorithm and figure match properly with the text, so the paper is highly readable.
- Table 1 and Figure 6 are different visualizations of the same results, and it seems that only one is needed.

[Punctuation, Grammar]
- It would be better to check the grammar of the paper again.

[Typo]
- In Algorithm 1, HER -> EHR

Author Response

(The authors gave the same response as above.)

Reviewer 3 Report

The article deals with secured health storage using blockchain technology. The article should be improved as it does not answer to serious questions:

  • why do you use blockchain (distributed ledger) in cooperation with centralized environment (EHR system server)
  • is there any advantage for blockchain in this case (what is the role of blockchain exactly).

Minor questions:

  • The proposed architecture is not presented clearly - contains serious flaws
  • The Fig. 1 should contain numbers for message flow
  • Algorithm design - entity running algorithms is not obvious from description
  • It is unusual to share privete key (algorithm1). Somethin in design goes wrong
  • Storage od password in Blockchain is at least questionable
  • who, how computes hash. The used type of blockchain and consensual algorithm is not discussed
  • How do you check that blockchain is safe - checking hash only detects change in block without hash modification. It does not ensure safety property

Author Response

(The authors gave the same response as above.)

Round 2

Reviewer 1 Report

The authors made the required modifications and improvements.

Author Response

I thank the reviewer for spending his valuable time reviewing my manuscript. As per the reviewer's previous recommendations, we have addressed all the comments.

Reviewer 2 Report

The revised version is not efficient to resolve all the review comments. The authors need to reconsider the following questions/comments and revise the paper again to solve them comprehensively.

Point 1. Verification of EHR

In the proposed method, patients and doctors each have their own blockchain of EHR, which needs to be verified. But the person who verifies the EHR will be himself, which gives a big security concern. There is no EHR verifier in the proposed architecture. How to solve this security problem of EHR integrity.

Point 2. BSF-HER system architecture

The proposed BSF-EHR system is a centralized system in a certain way. In the BSF-EHR system, each blockchain (patient, doctor, insurance agent) is uploaded to the BSF-EHR centralized server, and the EHR server is accessed by users to obtain the EHR. But normal blockchain system is a distributed system, where transaction data is created, verified, and stored by any distributed users. The authors should explain how the proposed BSF-EHR system works in a way of distributed system.

Point 3. Scalability problem

A scalability problem occurs in the proposed system. In the EHR blockchain in the proposed system, there is only one transaction in a HER block which makes scalability problem, where the block height will be very long in a short period. It will be difficult to manage and store the block data.

Point 4. Blockchain validation algorithm

In the proposed method, the validation of EHR blocks is performed only through hash comparison. This algorithm is not enough to validate block data integrity.

Point 5. Comparison of BSF-EHR with other related works

In Table I, the authors mentioned that BSF-EHR outperformed better than several related works. But the evidence of the better performance is not explained in detail, such as performance metrics and experiment details. More detail explanation is needed.

Point 6. BSF-EHR vs centralized storage

Although the author mentioned that BSF-EHR outperforms the centralized storage in terms of time consumption. But it is not clear that the proposed BSF-HER satisfies the security and privacy concerns which should be the most important contribution in the proposed system.

Point 7. Future work

The author mentioned about update of blockchain data as future work. This directly conflicts with the purpose of the blockchain mechanism. It is necessary to mention what data needs to be updated in a long chain of EHR blocks, which means that the blockchain mechanism is unnecessary for EHR.

Author Response

I thank the reviewer for spending his valuable time reviewing my manuscript. As per the reviewer's recommendations, we have addressed all the comments.

Reviewer 3 Report

The article has been improved, there is only one missing response to my precious review:

Is EHR centralized element (even using distributed DB etc)? If no, it should be incuded somehow in the article. If it is centralized element, I still do not see the benefit of the blockchain because you loose the one of CAP (CAP theorema) benefits. 

If it is centralized element, who and how is participating on the "mining" process. However I suggest not to use PoW algorithm, which is a bit old-fashioned.

I still suggest not to distribute "private key". As the name suggest, it is private. There are other relevant approaches how to share information without private key distribution.

Author Response

(The authors gave the same response as above.)

Round 3

Reviewer 2 Report

Most of my comments were addressed appropriately in the revised manuscript, and the structure and design of the BSF-EHR system has become clear to understand. The revised version is eligible to be accepted except the below some minor points.

Point1. Blockchain validation algorithm

In the previous proposed blockchain verification algorithm, blocks are validated only by comparing hash values. In the revised blockchain verification algorithm, a Bilinear Map comparison is added for EHR integrity verification. Detail explanations of the newly proposed Bilinear map are needed in the blockchain verification algorithm.

Point2. BSF-EHR system architecture

The overall structure and design of the BSF-EHR system was understood through the author's response, but the explanation for EHR system server and the blockchain structure of the patient or physician is not clear through this paper. In this paper, an additional description of the BSF-EHR system structure and a specific description of the overall flow are required.

Point3. BSF-EHR system design

The authors mentioned what functions each blockchain performs, but only the basic concept of the blockchain is practically mentioned. The role of blockchain and its benefit in the BSF-EHR system need to be explained in detail.

Author Response

As per the reviewer's comments, all the comments were addressed and point by point reference is uploaded as a separate word file.
